# An Improved Phenotype-Driven Tool for Rare Mendelian Variant Prioritization: Benchmarking Exomiser on Real Patient Whole-Exome Data

**DOI:** 10.3390/genes11040460

**Published:** 2020-04-23

**Authors:** Valentina Cipriani, Nikolas Pontikos, Gavin Arno, Panagiotis I. Sergouniotis, Eva Lenassi, Penpitcha Thawong, Daniel Danis, Michel Michaelides, Andrew R. Webster, Anthony T. Moore, Peter N. Robinson, Julius O.B. Jacobsen, Damian Smedley

**Affiliations:** 1William Harvey Research Institute, Queen Mary University of London, London EC1M 6BQ, UK; j.jacobsen@qmul.ac.uk (J.O.B.J.); d.smedley@qmul.ac.uk (D.S.); 2UCL Institute of Ophthalmology, University College London, London EC1V 9EL, UK; n.pontikos@ucl.ac.uk (N.P.); g.arno@ucl.ac.uk (G.A.); michel.michaelides@ucl.ac.uk (M.M.); andrew.webster@ucl.ac.uk (A.R.W.); Tony.Moore@ucsf.edu (A.T.M.); 3Moorfields Eye Hospital NHS Foundation Trust, London EC1V 2PD, UK; 4UCL Genetics Institute, University College London, London WC1E 6AA, UK; 5North East Thames Regional Genetics Laboratory, Great Ormond Street Hospital NHS Trust, London WC1N 3BH, UK; 6Manchester Royal Eye Hospital & University of Manchester, Manchester M13 9WL, UK; panagiotis.sergouniotis@manchester.ac.uk (P.I.S.); eva.lenassi@gmail.com (E.L.); 7Department of Medical Sciences, Medical Genetics Section, National Institute of Health, Ministry of Public Health, Nonthaburi 11000, Thailand; thawong.p@dmsc.mail.go.th; 8The Jackson Laboratory for Genomic Medicine, Farmington, CT 06032, USA; Daniel.Danis@jax.org (D.D.); Peter.Robinson@jax.org (P.N.R.); 9Ophthalmology Department, UCSF School of Medicine, San Francisco, CA 94143-0644, USA

**Keywords:** whole-exome sequencing, whole-genome sequencing, rare disease, variant prioritization, human phenotype ontology, phenotypic similarity, bioinformatics, inherited retinal disease

## Abstract

Next-generation sequencing has revolutionized rare disease diagnostics, but many patients remain without a molecular diagnosis, particularly because many candidate variants usually survive despite strict filtering. Exomiser was launched in 2014 as a Java tool that performs an integrative analysis of patients’ sequencing data and their phenotypes encoded with Human Phenotype Ontology (HPO) terms. It prioritizes variants by leveraging information on variant frequency, predicted pathogenicity, and gene-phenotype associations derived from human diseases, model organisms, and protein–protein interactions. Early published releases of Exomiser were able to prioritize disease-causative variants as top candidates in up to 97% of simulated whole-exomes. The size of the tested real patient datasets published so far are very limited. Here, we present the latest Exomiser version 12.0.1 with many new features. We assessed the performance using a set of 134 whole-exomes from patients with a range of rare retinal diseases and known molecular diagnosis. Using default settings, Exomiser ranked the correct diagnosed variants as the top candidate in 74% of the dataset and top 5 in 94%; not using the patients’ HPO profiles (i.e., variant-only analysis) decreased the performance to 3% and 27%, respectively. In conclusion, Exomiser is an effective support tool for rare Mendelian phenotype-driven variant prioritization.

## 1. Introduction

Despite the tremendous advances brought by next-generation sequencing to the field of rare Mendelian gene discovery and diagnostics, many challenges remain [1,2], and this is reflected in limited diagnostic yield. A recent review of the results of whole-exome sequencing (WES) in rare pediatric disorders reported variable diagnostic rates depending on the diagnosis. Positive results were reported in 17% for renal disease, 56% for non-syndromic deafness and inherited retinal disease (IRD), and 76% for ciliary dyskinesia [3]. Several limitations of high-throughput sequencing still exist that may impact the diagnostic yield. These include the yet incomplete coverage affecting especially short-read sequencing, the nonexistence of a de facto standard calling algorithm for copy-number variants (CNVs), or the challenges in filtering and interpreting short tandem repeats with no known disease association [4]. Another major reason why a disorder remains unsolved after undergoing next-generation sequencing is the complexity of the interpretation of the wealth of variants found, which is further hindered by the incomplete knowledge on gene functions [4]. For example, an individual WES experiment typically identifies tens of thousands of variants (when compared to the reference genome). Of these, roughly 10,000 may be predicted to result in nonsynonymous amino acid changes, to alter conserved splice sites, or to represent small insertions or deletions (indels). Strict frequency filtering is generally applied to discard common variants (seen in publicly available and/or in-house sequencing datasets) that would not explain a rare disease phenotype and/or those variants that would not fit the expected mode of inheritance. Nevertheless, additional criteria are usually needed to refine the number of possible candidate variants. On the other hand, although selecting variants that are extremely rare or absent in population-based reference datasets may be helpful in a first step of the analysis, using very low frequency cutoffs may remove clinically relevant variants and may lead to false negative findings, especially in hard to solve cases [5]. More complex statistical frameworks that account for disease prevalence, genetic and allelic heterogeneity, inheritance mode, penetrance, and sampling variance in reference datasets have been recently suggested for a more effective frequency-based variant filtering [5,6].

A growing number of in silico (freely available) computational tools have been developed to predict the pathogenicity of missense variants [7,8]. These include (i) function-prediction methods based on the likelihood of a given missense variant causing pathogenic changes in protein function (e.g., FATHMM [9], fitCons [10], LRT [11], MutationAssessor [12], MutationTaster [13], PolyPhen-2 [14], PROVEAN [15], SIFT [16], and VEST3 [17]), (ii) conservation methods that use multiple alignments to measure the degree of conservation at a given nucleotide site (e.g., GERP++ [18], phastCons [19], phyloP [20], and SiPhy [21]), and (iii) ensemble methods that integrate information from multiple component methods (e.g., CADD [22], DANN [23], Eigen [24], FATHMM-MKL [25], GenoCanyon [26], M-CAP [27], MetaLR [28], MetaSVM [28], REVEL [29], GAVIN [30], and MPC [31]. Other more recent methods concern a tailored use of deep neural networks (e.g., MVP [32] and PrimateAI [33]). Overall, there is a lack of agreement of the different methods in classifying variants as benign or disease-causing. A recent study thoroughly assessed the performance of 23 of these methods in evaluating 3 independent benchmarking datasets, including clinical variants from ClinVar database (https://www.ncbi.nlm.nih.gov/clinvar), somatic variants, and experimentally evaluated variants [8]. The results demonstrated that the specificities were lower than the sensitivities for most methods across datasets, suggesting the need for more stringent cutoff values to distinguish pathogenic variants. For example, using the ClinVar dataset, the specificities ranged from 35% to 90% (median, 65%) and the sensitivities ranged from 51% to 96% (median, 88%). Altogether, the study confirmed that using variant prioritization methods based purely on putative sequence variant pathogenicity is expected to generate many false positive candidates.

In order to overcome these difficulties, another generation of (freely available) tools have been developed that incorporate a rare disease patient’s phenotype into the interpretation of their sequencing data. These include eXtasy [34], BiERapp [35], Phen-Gen [36], Exomiser [37], Phevor [38], PhenoVar [39], PhenIX [40], OVA [41], Phenolyzer [42], wANNOVAR [43], OMIM Explorer [44], QueryOR [45], GenIO [46], DeepPVP [47], MutationDistiller [48], Phrank [49], Xrare [50], PhenoPro [51], and Phenoxome [52]. Such phenotype-driven prioritization tools leverage existing genotype to phenotype information from various databases in order to prioritize candidate variants in those genes that are likely to be more relevant to the patient’s phenotype. In other words, these methods try to automate many of the manual interpretation steps that clinicians and molecular geneticists usually perform to identify those genes that show connections with the phenotype of a patient. Although these methods vary in many methodological and technical aspects, they all require that the patient phenotypic information is encoded into terms from the Human Phenotype Ontology (HPO) [53,54,55,56] to make it readable and interpretable for computational analysis. The HPO is a flagship product of the Monarch Initiative (https://monarchinitiative.org/), a National Institutes of Health (NIH)-supported international consortium for semantic integration of biomedical and model organism data with the ultimate goal of improving biomedical research. The ontology currently contains over 13,000 terms describing human phenotypic abnormalities and over 156,000 annotations to hereditary diseases and has become the de facto standard for deep phenotyping in the field of rare disease [53].

Exomiser was among the first bioinformatics tools of its kind, published in 2014 [37] as a freely available Java program. It requires as input (i) a variant call format (VCF) file with the called variants of a rare disease patient (or optionally a multi-sample VCF and pedigree (PED) file if family members have also been sequenced) and (ii) a set of HPO terms to describe the corresponding patient’s phenotype. A range of user-defined variant filtering criteria can be applied based on JANNOVAR [57] functional annotation, frequency, and expected inheritance pattern. The filtered variants are then prioritized according to a variant score based on their rarity and in silico algorithm-predicted pathogenicity, combined with a gene-specific phenotype score. In the original release [37], the phenotype score was calculated based on the semantic similarity (obtained via the PhenoDigm algorithm [58]) between the user-provided HPO-encoded patient’s phenotype and the phenotypic annotations of orthologous genes in mouse model organisms. Subsequently, this has been extended to cover also the phenotypic annotations of genes in known human diseases, orthologs in zebrafish model organisms, and phenotypes of protein–protein associated neighbors [59,60] (Figure 1). Furthermore, Exomiser now allows both coding and noncoding variant prioritization from whole-genome sequencing data [61].

Exomiser was able to prioritize causative variants as top candidates in up to 97% of 10,000 simulated rare disease whole-exomes (based on 28,516 known disease-causing mutations from the human gene mutation database (HGMD, http://www.hgmd.org) and on 1092 whole-exome VCF files from the 1000 Genomes Project) [59]. Software performance statistics on real patient whole-exome datasets published so far remain very limited; eleven previously identified disease-causing variants from real patient data were ranked in the top 7 hits of their respective lists by Exomiser and 6 of 11 were ranked first [59]. Since its first published releases, Exomiser has been updated multiple times with a number of improved and additional features. The aims of this report were i) to present the latest release of Exomiser (version 12.0.1) with a description of the main newly added features and ii) to show the corresponding software performance statistics on a real patient whole-exome dataset from 134 individuals affected by rare IRD who had previously received a molecular diagnosis based on the analysis of their NGS data. Remarkably, Exomiser effectively prioritized the correct disease-causing variants among the top 5 candidates for most of the patients (i.e., 94%). We recommend adopting Exomiser to guide and accelerate the typically complex and time-consuming, multidisciplinary work required to prioritize candidate variants and to provide molecular diagnoses in rare Mendelian disease-sequencing efforts.

## 2. Materials and Methods

### 2.1. Real Patient Whole-Exome Dataset with Known Molecular Diagnosis

We evaluated Exomiser software performance using a real patient whole-exome dataset with known molecular diagnosis obtained from a number of next-generation sequencing studies that were conducted at Moorfields Eye Hospital and the University College London (UCL) Institute of Ophthalmology (London, UK) approximately over four years (2011–2015) [62]. The dataset was extracted from the overall patient cohort of 340 individuals that had been diagnosed with IRD by a consultant ophthalmologist and that had undergone WES. All patients had been sequenced as singletons, except for six probands where one or both parents were also sequenced. The patient cohort was enriched for rare genetic causes of IRD (many of which have been published over the years). Cases were selected for their likelihood of harbouring a novel discovery based on many features, including family consanguinity/number of affected siblings in recessive families, prior negative testing for the Asper Ophthalmics chip (Asper Biogene, Tartu, Estonia), candidate gene sequencing or the Manchester 105-gene diagnostic panel, unusual phenotype/clinical history, prior single nucleotide polymorphism linkage/homozygosity mapping, absence of typical clinical features of *ABCA4*-retinopathy, or other (near) monogenic diseases (e.g., X-linked retinoschisis, *RS1*, or choroideremia, *CHM*). A total of 134 IRD patients that had received a molecular diagnosis (i.e., “solved” patients) at the time of the data extraction based on single nucleotide variants (SNVs) and/or indels (as present in their respective VCF files) were included in the Exomiser software performance assessment. Twelve additional patients with a molecular diagnosis that involved CNVs could not be used due to the current software inability of incorporating CNVs into the analysis. The molecular diagnosis of the IRD patients had been based on the analysis and interpretation of the patient WES data performed by various clinicians, scientists, and analysts without using Exomiser. Where possible, diagnosed variants, including compound heterozygous variants, had been validated in an accredited clinical diagnostic laboratory. Familial segregation analysis had been performed where appropriate to reach a diagnostic conclusion. The study protocols adhered to the tenets of the Declaration of Helsinki and received approval from the local ethics committee (MOOA1016, 13/EE/0325). Written informed consent was obtained from all participants, or their parents, before their inclusion in the study.

Table 1 shows the frequency distribution of the clinical diagnosis (assigned before performing WES), with the majority of the patients being diagnosed with retinitis pigmentosa (27%), Leber congenital amaurosis (19%), macular dystrophy (12%), and cone-rod dystrophy (10%). Following WES, for most of the patients, the molecular diagnosis was based either on a homozygous genotype (72 out of 134) or a compound heterozygous genotype (39) (Table 2). The full list of disease-associated genes and variants for the 134 IRD solved patients is shown in Appendix A. Mutations in 60 genes were identified to be disease-causing among the 134 IRD patients (Appendix A), with variants in *USH2A* accounting for 7% of the dataset, variants in *RPGR* and *ABCA4* accounting for 6%, and variants in *GUCY2D* accounting for 5%. The dataset exhibited both genetic and phenotypic heterogeneity (Appendix A). For example, variants in four different diagnosed genes (*ATF6, CNGB3, GNAT2*, and *POC1B*) were associated to the same clinical diagnosis achromatopsia (genetic heterogeneity), while variants in *CRB1* were found in patients who had been diagnosed with Leber congenital amaurosis and retinitis pigmentosa (phenotypic heterogeneity). In the cases of *CLN3, COL18A1*, *HPS6* and *SRD5A3*, the corresponding patients had been initially diagnosed with non-syndromic IRD and, after further clinical examination, following the analysis of their WES data, were assigned to a syndromic IRD diagnosis (Batten disease, Knobloch syndrome, Hermansky-Pudlak syndrome, and *SRD5A3*-related congenital disorder of glycosylation, respectively). The set of disease-associated genes in this cohort included both established and more recently identified IRD genes (for a continuously updated list of IRD-associated genes, see https://sph.uth.edu/retnet/).

### 2.2. Human Phenotype Ontology (HPO)-Encoded Clinical Diagnoses

In order to overcome the laborious process of acquiring patient-specific deep phenotypes to translate into HPO terms, each clinical diagnosis was assigned a parsimonious, fixed list of most representative HPO terms (from one single term to six terms) by three ophthalmologists with expertise in IRD diagnosis (Appendix A). The graphic visualization of the HPO terms for Leber congenital amaurosis as subgraphs of the full ontology are presented in Figure 2. Leber congenital amaurosis was annotated to four HPO terms, i.e., *retinal dystrophy* (HP:0000556), *visual impairment* (HP:0007758), *undetectable electroretinogram* (HP:0000550), and *nystagmus* (HP:0000639), and, implicitly, to the parental terms in the ontology, i.e., *phenotypic abnormality, eye,* and *eye morphology* for *retinal dystrophy* and *phenotypic abnormality, eye,* and *eye physiology* for the other 3 terms. The HPO graphic visualization for all the clinical diagnoses observed in the IRD patient dataset is depicted in Appendix A.

### 2.3. Exomiser Software

We tested the latest release of Exomiser (version 12.0.1, downloaded in August 2019) available at https://github.com/exomiser/Exomiser. A number of new features have been added since the early published software releases [37,59] and are described below.

The functional annotation of variants (handled by the JANNOVAR library [57]) can now be done using any of UCSC, RefSeq, or Ensembl KnownGene transcript definitions, together with either hg19 or hg38 genomic coordinates. Analysis of mitochondrial data is now possible (assuming mitochondrial variants are called in the input VCF file, they will be filtered and interpreted in the same manner as nuclear variants). In addition to the 1000 Genomes Project [63], the National Heart, Lung, and Blood Institute Grand Opportunity (NHLBI GO) Exome Sequencing Project (ESP, http://evs.gs.washington.edu/EVS), and the Exome Aggregation Consortium (ExAC) [64] datasets used in earlier software releases, variant frequency data used in the filtering step are now taken from additional large sequencing studies, including Trans-Omics for Precision Medicine (TOPMed) (https://www.nhlbiwgs.org), UK10K [65], and Genome Aggregation Database (gnomAD) [66]. Together with publicly available datasets, the user can additionally specify an in-house dataset of variant allele frequency (LOCAL) for variant filtering. The maximum minor allele frequency (MAF) seen across all the selected datasets is used for comparison with the chosen allele frequency cutoff; the allele frequency cutoffs can now be tailored to each selected mode of inheritance (autosomal dominant, AD; autosomal recessive, AR; X-linked dominant, XD; X-linked recessive, XR; and MT) and the user can decide to run the analysis for multiple modes of inheritance simultaneously. For AR and XR, the variant allele frequency cutoffs can be specified for homozygous and compound heterozygous genotypes separately.

After variant filtering by Phred quality score (option *qualityFilter*), type of variant (option *variantEffectFilter*), and allele frequency by mode of inheritance (options *FrequencyFilter*, *frequencySources* and *inheritanceModes*), Exomiser assigns each remaining filtered variant a “variant score” (from 0 for benign to 1 for pathogenic) that is the product of the variant frequency score and the variant pathogenicity score (Figure 1). The variant frequency score between 0 and 1 is now calculated as 1.13533 − (0.13533 · *e*^100*f*^) (where *f* is the maximum MAF seen in the selected variant databases in the filtering settings) and is set to 0 if *f* is > 2 and to 1 if *f* is equal to 0. The variant pathogenicity score is defined as the maximum of the pathogenicity scores selected by the user (after transforming and/or normalizing them to lie between 0 for benign and 1 for pathogenic). In the earlier published software releases [37,59], the user could select the pathogenicity scores (option *pathogenicitySources*) among PolyPhen-2 [14], MutationTaster [13], and SIFT [16]; the ensemble method CADD [22,67] was added later, followed by the more recent ensemble pathogenicity scores M_CAP [27], REVEL [29], and MPC [31], together with the latest deep neural network methods MVP [32] and PrimateAI [33], all available from software version 12.0.0. For frameshift, nonsense, canonical splice acceptor/donor, stop-loss, and start-loss variants, the pathogenicity scores are fixed and have been increased from 0.95 to the maximum score of 1 to reflect the clinical interpretation of these variant consequences; splice region variants are assigned score 0.8. When no pathogenicity data for a certain allele is present in the specified pathogenicity sources, Exomiser uses a preset value based on the most pathogenic, predicted variant consequence for all transcripts overlapping that genomic position. Another latest feature enables flagging variants on a whitelist and bypassing of the *variantEffectFilter* and *frequencyFilter* options. By default, the whitelist consists of ClinVar (https://www.ncbi.nlm.nih.gov/clinvar) variants listed as “Pathogenic” or “Likely_pathogenic” and with a review status of “criteria provided, single submitter” or better [68]. Indeed, the user can edit the variant whitelist (i.e., an openly available tabix indexed file) by adding any variants they want to keep, bypassing any filtering settings. Finally, in the case of a candidate compound heterozygous genotype, the final variant score is obtained from the average of the two corresponding variant scores.

As in earlier software releases, if the *hiPhivePrioritiser* option is enabled, the variant prioritization step is active and each gene containing any filtered variant(s) receives a “phenotype score” (from 0 to 1) (Figure 1). The gene-specific phenotype score is based on the best semantic similarity (calculated via the PhenoDigm algorithm [58]) between the user-provided patient’s phenotype (encoded as a set of HPO terms) and the phenotypic annotation of any known gene-associated phenotypes in: (i) a human disease (as curated in OMIM [69] and Orphanet [70]), (ii) a mouse model (as curated in the Mouse Genome Informatics (MGI) [71] and International Mouse Phenotypic Consortium databases; IMPC, www.mousephenotype.org), (iii) a zebrafish model (as curated in the Zebrafish Model Organism Database, ZFIN [72]) for the candidate gene; or iv) a human/mouse/zebrafish model for a neighboring gene in the interactome (obtained via STRING [73] analysis). If the *omimPrioritiser* option is enabled, in case a phenotype match with a known human disease gene is found, Exomiser checks whether the patient’s candidate genotype is consistent with the reported OMIM disease mode of inheritance; if there is inconsistency, the corresponding phenotype score is halved. Finally, the variants with the highest variant score within each candidate gene are flagged as “contributing” variants and their corresponding variant score is combined with the gene-specific phenotype score into the Exomiser score (from 0 to 1). The Exomiser score is obtained using a logistic regression classifier on a training set of 10,000 disease and 10,000 benign variants with a ten-fold cross-validation as previously described [37,59] and further optimized in latest software releases (Figure 1).

Other recent changes include the split of the internal Exomiser database into a variant and a phenotypic database to allow for more frequent and smaller updates; the variant database now stores information on over 300 million variants for each assembly in a new highly compressed data format that enables much smaller on-disk footprint with minimal loss of read performance.

### 2.4. Software Analysis Settings

Exomiser was run on each single-patient VCF file in combination with their respective HPO-encoded clinical diagnosis; the IRD patient dataset comprised only unrelated patients and therefore no PED file was required. We repeated the analysis on the same patient dataset under a variety of settings intended: to mimic clinical diagnostic conditions, to allow comparison with previously published results on simulated data, to assess the value of integrating the HPO patients’ profiles in the variant prioritization analysis, and to gain insights into some of the newly added software features. Ensembl transcript annotation was used across all the analysis settings, together with the same options and values for the filtering step (Table 3). The allele frequency cutoffs per each mode of inheritance were set to match the filtering criteria of the bioinformatics pipeline that had been used to solve the cases. As an optional LOCAL in-house dataset for variant filtering, we used UCLex [74], an exome database of about 4500 whole-exomes collected at University College London (UK) from various research groups since 2012, to which the IRD patients included in this analysis belong to. The performance of Exomiser on the IRD patient dataset was assessed using the following eight analysis settings (Table 3):

1) DEFAULT: the Exomiser score was obtained from the variant score based on allele frequency (as defined in the filtering step) and the original pathogenicity algorithms PolyPhen-2 [14], MutationTaster [13], and SIFT [75] using

*pathogenicitySources*: [POLYPHEN, MUTATION_TASTER, SIFT],

plus the gene-specific phenotype score, using

hiPhivePrioritiser: {},

and

omimPrioritiser: {}.

2) VAR-ONLY: as in analysis setting 1) but using only the variant score, i.e., *hiPhivePrioritiser* and *omimPrioritiser* were disabled (to gain insights into any added value of using the phenotype score in the Exomiser calculation);

3)–8) CADD/REVEL/MPC/M_CAP/MVP/PRIMATE-AI: as in analysis setting 1) using both the variant score and the phenotype score but making use of only one of the newly added pathogenicity algorithms at a time to calculate the variant score, e.g., *pathogenicitySources*: [CADD] (to gain insights into the behavior of the newly added pathogenicity algorithms).

Exomiser was run on the Queen Mary University’s Apocrita HPC facility (supported by QMUL Research-IT (http://doi.org/10.5281/zenodo.438045) using the following command-line per each single-sample analysis and each of the eight different analysis YML files shown in Table 3:

java -Xms2g -Xmx4g -jar exomiser-cli-12.0.1.jar --analysis *AnalysisFilename*.yml

with Java version 1.8, Exomiser variant and phenotype data version 1902, CADD data version 1.4 (Genome build GRCh37/hg19; https://cadd.gs.washington.edu/).

### 2.5. Software Performance Evaluation and Statistical Analysis

To assess the performance of Exomiser on the IRD patient dataset, we noted the rank at which the software outputs the correct diagnosed disease variant(s) in each single-patient output (.tsv) file for the corresponding solved mode of inheritance (i.e., AR for the homozygous and compound-heterozygous solved genotypes, AD for the heterozygote solved genotypes, and XR for the hemizygous solved genotypes with variants on the X chromosome) (Appendix A). Equally scoring gene-variant observations (ties) were resolved by assigning the corresponding average rank. The pairwise agreement among the Exomiser ranking results between DEFAULT and VAR-ONLY settings and between DEFAULT and each of the six single pathogenicity algorithm-based settings, i.e., CADD, REVEL, MPC, M_CAP, MVP, or PRIMATE-AI, were evaluated using the Wilcoxon signed rank test for paired data. We also categorized the ranking results for the causative variants into five mutually exclusive disease-causing ranking bins: “Top”, “2–5”, “6–10”, “>10”, and “Filtered out/Not prioritized” (the latter being any disease-causing variant(s): failed to be kept in during the filtering step/not selected as “contributing” variants to the final Exomiser score during the prioritization step) and calculated the corresponding Cohen’s kappa coefficient (κ). Cohen’s kappa values were interpreted according to Landis and Koch’s guidelines (i.e., κ < 0.00 as “poor” agreement, 0.00–0.20 as “slight”, 0.21–0.40 as “fair”, 0.41–0.60 as “moderate”, 0.61–0.80 as “substantial”, and 0.81–1 as “almost perfect” agreement) [76]. The nonparametric Stuart–Maxwell test was then used to assess the marginal homogeneity of the five ranking bins simultaneously. All the statistical analyses were conducted using Stata software (version 14.2; StataCorp LLC, College Station, TX, USA) *signrank*, *symmetry*, and *kap* commands; the HPO graphic visualization was carried out using R version 3.5.3 within Rstudio version 1.1.442 (*ontologyIndex* and *ontologyPlot* packages [77]).

### 2.6. Whole-Exome Sequencing Data

Genomic DNA was purified from peripheral blood mononuclear cells for WES. WES was performed independently or as part of the National Institute for Health Research BioResource—Rare Disease study (NIHR BR-RD) as previously described [78]. Samples were analyzed as part of the UCLex WES dataset using the Phenopolis bioinformatics pipeline [74]. Reads were aligned to the hg19 human reference sequence (build GRCh37) with NovoAlign (version 3.02.08; Novocraft Technologies, Petaling Jaya, Malaysia). The aligned reads were sorted by base pair position and duplicates were marked using NovoSort (version 1.03.05; Novocraft Technologies). Discordant reads were identified using SAMBLASTER (version 0.1.25) [79]. Variant calling and recalibration were performed using the Genome Analysis Tool Kit (GATK) (version 3.5) (https://www.ncbi.nlm.nih.gov/pubmed/20644199). As per the GATK Best Practices, variant calling from the aligned reads was performed using *HaplotypeCaller* and joint variant calling was performed using *GenotypeGVCFs*. Variant Quality Score Recalibration (VQSR) was performed separately for SNVs and indels to estimate the variant quality score log-odds (VQSLOD) and the FILTER values per variant.

## 3. Results

### 3.1. Exomiser Performance on the Inherited Retinal Disease (IRD) Patient Dataset

We benchmarked Exomiser version 12.0.1 on a dataset of 134 whole-exomes from IRD patients who had received a molecular diagnosis which included SNVs and/or indels. After applying the filtering step (Figure 1) (with identical options and values used across all the eight analysis settings, Table 3), each patient’s exome was found to carry an average number of 474 variants (range 229–1276) in an average number of 358 genes (range 193–925). The correct diagnosed causative variants (Appendix A) were kept within the filtered variant dataset for all patients, apart from patient P85 (clinically diagnosed with retinal dystrophy) and patient P94 (clinically diagnosed with retinitis pigmentosa) (“Filtered out”, Table 4 and Figure 3). Patient P85 had been solved with a homozygous frameshift truncation variant (Glu211Aspfs*13) in *LCA5* gene; this homozygous variant had been detected only via manual inspection of the sequencing data (at a low covered region) via the Integrative Genomics Viewer (IGV) and further experiments. The variant had been instead interpreted as heterozygous by the calling variant pipeline and therefore did not pass the Exomiser AR mode of inheritance filter. Similarly, for patient P94, the diagnosed variant had been erroneously called as heterozygous in the VCF file and did not pass the Exomiser XR mode of inheritance filter.

Using the DEFAULT analysis settings (Table 3), Exomiser was able to assign the correct causative variants first rank for 99 (74%) IRD patients; the diagnosed mutations were ranked top 5 and top 10 in 94% and 96% of the patients’ exomes, respectively; the median and mean ranks of the causative variants were 1 and 2.1, respectively (range: 1–42) (Table 4 and Figure 3). For patient P126 (clinically diagnosed with Usher syndrome type I and molecularly diagnosed as compound heterozygote with variants in *ADGRV1* gene, Appendix A), the correct diagnosed gene was ranked first, but only one of the two diagnosed heterozygous variants was selected as “contributing” to the final Exomiser score (“Not prioritized”, Table 4 and Figure 3). The best semantic similarities that determined the gene-specific phenotype scores were obtained from the human disease databases (i.e., OMIM/Orphanet) for 125 (94.7%) exomes out of the 132 exomes for which the correct variants (and corresponding genes) were not filtered out. The HPO profiles used to annotate the patients’ clinical diagnoses either matched (to a certain extent) the HPO annotation in OMIM/Orphanet for the same disorder of the patient or a phenotypically similar one (e.g., retinitis pigmentosa and Bardet-Biedl syndrome 1, OMIM:209900, or cone-rod dystrophy and Joubert syndrome with ocular defect, ORPHA:220493). As to the remaining of the IRD dataset, the best phenotype matches were observed with a mouse model for 3 (2.3%) exomes (with the correct genes ranked 1, 1, and 2) and with a zebrafish model for 1 (0.7%) exome (with the correct gene ranked 5), and based on data from the protein–protein database for 3 (2.3%) exomes (with the correct genes ranked 5, 7, and 12).

In order to gain insights into the added value of using the phenotype score in the Exomiser score calculation (Figure 1), we run the VAR-ONLY analysis using only the variant score (with the same pathogenicity algorithms as per the DEFAULT settings, i.e., Polyphen-2, MutationTaster, and SIFT) (Table 3). The correct causative variants were ranked first only for 4 out of 134 patients; while in 27% and 55% of the patients’ exomes the diagnosed mutations were ranked top 5 and top 10, respectively, the median and mean ranks of the causative variant(s) were 9.5 and 10.8, respectively (range: 1–60.5) (Table 4 and Figure 3). The percentage agreement between DEFAULT and VAR-ONLY settings was only 8% (“poor” κ = −0.013); overall, the DEFAULT settings significantly outperformed the VAR-ONLY settings (Wilcoxon signed rank test, *p*-value = 1.0 × 10^−20^) (Table 5 and Figure 3).

We also run Exomiser using only one of the newly added six pathogenicity algorithms (i.e., either CADD, REVEL, MPC, M_CAP, MVP, or PRIMATE-AI) to calculate the variant score, together with the phenotype score to obtain the final Exomiser score as per in the DEFAULT settings (Table 3). The ensemble CADD and REVEL settings were both able to rank the correct disease-causing variants in 72% of the IRD dataset; similarly, the deep neural network MVP and PRIMATE_AI settings assigned the diagnosed mutations first rank in 77% and 73% of the dataset (Table 4 and Figure 3). Overall, none of these four single pathogenicity algorithm-based settings showed a statistically different performance from the DEFAULT settings (Wilcoxon signed rank test, *p*-value = 0.195, 0.511, 0.400, and 0.556, for the comparison of DEFAULT with CADD, REVEL, MVP, and PRIMATE-AI, respectively) with high percentage agreements (Table 5 and Figure 3). The pairwise DEFAULT-CADD comparison showed the highest percentage agreement (92%) with a “substantial” κ value equal to 0.80. On the other hand, despite moderate agreement, the DEFAULT settings mostly outperformed the MPC and M_CAP settings (Wilcoxon signed rank test, *p*-value = 7.4 × 10^−6^ and 0.001, with “fair” κ values equal to 0.24 and 0.27, respectively), where the disease-causing variants were ranked first in 56% and 63% of the IRD patient dataset, respectively (Figure 3).

### 3.2. Examples of Exomiser Results on the IRD Patient Dataset

Below, we describe some representative examples of the Exomiser results on the IRD patient dataset along with the main Exomiser output features.

Figure 4 shows a screenshot of the HTML output file (from the DEFAULT analysis) for patient P127 who was clinically diagnosed with Usher syndrome type II and molecularly diagnosed as compound heterozygote with frameshift elongation c.920_923dup:p.(His308Glnfs*16) and inframe deletion c.3832_3834del:p.(Leu1278del) in *USH2A* gene (Appendix A). The correct causative variants were assigned first rank with an Exomiser score equal to 0.965, obtained from a phenotype score equal to 0.780 and a variant score equal to 0.924. The phenotype score was derived from the phenotype match of the HPO profile used for Usher syndrome type II patients (Appendix A) and OMIM:276901 (i.e., Usher syndrome, type 2A) as the most similar phenotypic annotation of *USH2A* found in OMIM via the PhenoDigm algorithm; Exomiser also reported the phenotype matches (with lower phenotype scores) found for a mouse mutant (0.496) and a zebrafish mutant (0.346) involving *USH2A* as well as a human disease, mouse mutant, and zebrafish mutant of the *PDZD7* neighboring gene (0.538) found in the *USH2A* interactome via STRING analysis. The variant score 0.924 was obtained as the average of the variant scores of the frameshift elongation variant (1.0) and the inframe deletion variant (0.849) that formed the “contributing” compound heterozygous genotype; the frameshift elongation variant was assigned a variant score equal to 1.0 by default, as it belongs to the internal ClinVar whitelist. Exomiser also reported two other variants that passed the filtering step (with lower variant scores than the frameshift elongation and inframe deletion “contributing” variants) (Figure 4). Appendix A depicts an analogous screenshot of the HTML output file (from the DEFAULT analysis) for patient P42 who was clinically diagnosed with cone-rod dystrophy and molecularly diagnosed with splice acceptor variant c.3191-1G>T:p.? and missense variant c.2861A>C:p.(Tyr954Ser) in *ABCA4*. Exomiser assigned the correct causative variants first rank with a score equal to 0.998, obtained from a phenotype score equal to 1.0 and a variant score equal to 0.999. In this case, Exomiser found a perfect phenotype match between the HPO profile used for cone-rod dystrophy patients (Appendix A) and the phenotypic annotation for OMIM:604116 (i.e., Cone-rod dystrophy 3). Phenotype matches with lower phenotype scores were also reported for a mouse mutant (0.571) involving *ABCA4* and a human disease (i.e., Macular dystrophy, vitelliform, 4) associated with the neighboring *IMPG1* gene (0.570) in the *ABCA4* interactome. The variant score 0.999 was obtained as the average of the variant scores of the two “contributing” variants that formed the compound heterozygous genotype, i.e., 0.999 for the splice acceptor variant (predicted to be pathogenic by MutationTaster and with maximum MAF equal to 0.0097% detected in the LOCAL database) and 0.999 for the missense variant (with highest pathogenic score equal to 1 as predicted by MutationTaster and maximum MAF equal to 0.0097% detected in the LOCAL database).

Interestingly, the correct disease-causing variants in *CLN3, COL18A1*, *HPS6*, and *SRD5A3*, for which the corresponding patients had received a new, more syndromic diagnosis following WES (patients P38, P41, P44, and P50; Appendix A), were all assigned either first or second rank (using the DEFAULT analysis settings) (Appendix A). That was the result of high variant scores (0.997, 1.0, 1.0, and 1.0, respectively) coupled with relatively high phenotype scores (0.798, 0.656, 0.598, and 0.905, respectively). Remarkably, all the phenotype scores were derived from the phenotype match of the HPO-encoded patient’s initial diagnosis with the correct (new) diagnosis in OMIM, i.e., OMIM:614075 (Hermansky-Pudlak syndrome 6) associated with *HPS6*, OMIM:267750 (Knobloch syndrome, type 1) associated with *COL18A1*, OMIM:612379 (Congenital disorder of glycosylation, type Iq) associated with *SRD5A3*, and OMIM:204200 (Ceroid lipofuscinosis, neuronal, 3; Batten disease) associated with *CLN3*, respectively.

For five patients (4%), the correct disease-causing variants were ranked outside the top 5 candidates (using the DEFAULT analysis settings): patient P43 (Leber congenital amaurosis; *DHX38*) with rank 7, patient P4 (achromatopsia; *POC1B*) with rank 9, patient P27 (congenital stationary night blindness; *TRNT1*) with rank 12, patient P80 (Stickler syndrome; *COL9A3*) with rank 39, and patient P118 (retinitis pigmentosa; *RP1L1*) with rank 42 (Appendix A and Appendix A). The low-performing ranking seemed to derive from the relatively low phenotype scores, i.e., 0.502, 0.500, 0.378, and 0.405 for patients P43, P27, P80, and P118, respectively, while all the variant scores were relatively high, i.e., 0.997, 0.989, 0.997, and 0.997, respectively, with the exception of patient P4 where a moderate phenotype score (0.587) is coupled with a moderate variant score (0.848).

*DHX38* had been first associated to IRD (OMIM:618220) with homozygous missense variant c.995G>A:p.(Gly332Asp) found in four affected sibs from a consanguineous Pakistani family with early-onset retinitis pigmentosa and macular coloboma [80] and, more recently, has been confirmed to cause early-onset retinitis pigmentosa in two consanguineous Pakistani families with early-onset retinitis pigmentosa [81] with same homozygous missense variant c.971G>A:p.(Arg324Gln) also seen in patient P43 (Leber congenital amaurosis; rank 7) from the IRD dataset in this analysis. Exomiser did not identify any known gene-phenotype annotations in human disease (Appendix A) as the current version of the Exomiser phenotype database does not contain yet the OMIM:618220 phenotypic annotation for *DHX38*. Nevertheless, Exomiser detected some phenotypic similarity (0.502) of patient P43 with the OMIM:231550 (Achalasia-addisonianism-alacrimia syndrome) phenotypic annotation of the neighboring *AAAS* gene in the *DHX38* interactome. For patient P4 (achromatopsia; *POC1B*; rank 9), one of the two patient’s HPO profile terms, i.e., “Achromatopsia” (HP:0011516), was deemed to be moderately similar (0.587) with the phenotypic annotation “Abnormality of color vision” (HP:0000551) in OMIM:615973 (Cone-rod dystrophy 20; *POC1B*); the other patient’s HPO profile term, i.e., “Abnormal light-adapted electroretinogram” (HP:0008275), did not match any HPO term in OMIM:615973. In addition to that, for the diagnosed *POC1B* inframe deletion variant c.130_138del:p.(Ile44_Leu46del), no pathogenic data were available and a preset value of 0.85 was used, resulting in a moderate variant score (0.848). Similarly, for patient P27 (congenital stationary night blindness; *TRNT1*; rank 12), the phenotypic similarity of the patient with OMIM:616959 (Retinitis pigmentosa and erythrocytic microcytosis) was modest (0.476), and the best phenotypic similarity, still moderate (0.500), was found with the phenotypic annotation of the neighboring *RP9* gene (associated with Retinitis pigmentosa 9) in the *TRNT1* interactome. Finally, for patient P80 (*COL9A3*) and patient P118 (*RP1L1*), Exomiser found that the observed AR mode of inheritance of the “contributing” variants were incompatible with the reported OMIM mode of inheritance of the corresponding best phenotype match (AD OMIM:600969 and AD OMIM:613587, respectively). The fact that *COL9A3* is also annotated as AR in Orphanet (ORPHA:250984, Stickler syndrome) is not taken into account in the current implementation of the *omimPrioritiser* option. As a result of that, the otherwise high phenotype scores were halved as per Exomiser settings (from 0.755 to 0.378 and from 0.809 to 0.405, respectively).

## 4. Discussion

We used the latest version of the bioinformatics tool Exomiser to perform phenotype-driven variant prioritization analysis of whole-exome data to evaluate its performance in molecular diagnosis for rare Mendelian disease. Earlier published software versions had been tested primarily on simulated whole-exome data [37,59]. In contrast, we assessed the software performance using a real patient, whole-exome dataset from 134 patients affected by IRD with a known molecular diagnosis. Exomiser ranked the correct diagnosed variants as the top candidate in three out of four whole-exomes in the IRD dataset (using the DEFAULT analysis settings, Table 3) and within the top 5 candidates in up to 94% of the dataset (Table 4 and Figure 3).

At the core of the Exomiser analysis framework, there is the use of a phenotype score that evaluates how similar an HPO-encoded rare disease patient’s phenotype is with any known gene-phenotype annotations in human, mouse, and zebrafish. This is in addition to standard variant prioritization based on variant rarity and predicted pathogenicity (Figure 1). Running the analysis without making use of the Exomiser phenotype score (VAR-ONLY analysis settings, Table 3) showed a substantial decrease in performance with an unsatisfactory disease-causing variant ranking: the correct diagnosed variants were scored as top candidate in only 3% of the dataset and up to 27% within the top 5 candidates. The success of Exomiser in prioritizing disease-causing variants is dependent on accurate HPO terms that encode the patient’s phenotype, and this may be challenging in a busy clinical context. Our method of encoding the patient’s clinical diagnoses into a fixed list of HPO terms proved to be effective; however, many tools and platforms now exist that facilitate the efficient collection of patients’ HPO-encoded clinical phenotypes and/or automated extraction of human disease phenotypes from free text clinical notes as well as electronic health records [82,83,84,85].

Given the nature of the solved IRD exome dataset, with most diagnosed genes being either established retinal genes or published over the recent years, we expected the human disease databases contributing the most to the gene-specific phenotype scoring. For 125 (95%) exomes out of the 132 exomes for which the correct variants (and corresponding genes) were not filtered out (using the DEFAULT settings, Table 3 and Table 4), the best phenotype matches were obtained with an OMIM/Orphanet human disorder, either the same disorder of the patient or a phenotypically similar one (the latter case showing the flexibility of a semantic similarity method over the use of a fixed panel of candidate genes/diseases). Nevertheless, for the remaining 5% of the exomes where the best phenotype matches were based on mouse/zebrafish model or protein–protein interaction data, the correct diagnosed genes were still ranked rather effectively (i.e., 1, 1, 2, 5, 5, 7, and 12). For example, the diagnosed variant in *DHX38* seen in patient P43 (Appendix A and Appendix A) was assigned rank 7, despite this recently published disease-gene association not yet being included in the underlying human disease databases. Furthermore, we run a new analysis with same settings as per DEFAULT, but without using the human disease databases for the phenotype scoring. Despite the expected reduced performance in comparison with the full DEFAULT analysis, we observed a good software performance: Exomiser was able to assign the correct causative variants first rank for 82 (61%) IRD patients; the diagnosed mutations were ranked top 5 and top 10 in 80% and 87% of the patients’ exomes, respectively, with the median and mean ranks of the causative variants equal to 1 and 5.9, respectively (range: 1–69). This clearly shows the ability of Exomiser to assist the discovery of novel disease genes due to the incorporation of model organism and protein–protein interaction data.

The original benchmarking of Exomiser performed on 10,000 simulated rare disease whole-exomes showed that up to 97% of the data received the top scoring hit for the correct causative variants [59]. Despite the fact that a fair comparison is not possible due to the different software releases, it can be noted that the performance of Exomiser reduced by about 24% (for the first rank, using the DEFAULT analysis settings; Table 3, Table 4, and Figure 3) when using the real patient IRD dataset presented in this analysis. That may somewhat be expected due to the likely inherently more complex nature of real patient data; also, to describe the phenotypes of the simulated exomes, the authors used perfect HPO term matching derived from OMIM for each disease, and that might have affected the good performance. On the same line, given that Exomiser was originally benchmarked using PolyPhen-2, MutationTaster, and SIFT as pathogenicity sources, our software performance estimates based on the use of each of the newly added pathogenicity algorithms one at a time (i.e., CADD/REVEL/MPC/M_CAP/MVP/PRIMATE-AI analysis settings, Table 3) should be regarded as an exploratory assessment of how well the new pathogenicity scores perform with the existing Exomiser framework rather than a direct comparison of how effective each scoring system is.

Notably, for all the four patients that had received a new (more syndromic) diagnosis based on the findings from their WES data and further clinical examination (patients P38, P41, P44, and P50; Appendix A), Exomiser not only provided an excellent prioritization of the correct diagnosed variants (i.e., first rank for three patients and second rank for one patient, using the DEFAULT analysis settings, Table 3) but also, remarkably, unveiled the phenotype matches with the correct new diagnoses. These findings show the ability of Exomiser to support unbiased and efficient differential diagnostics based on the automated interpretation of sequencing data in the context of known human disease gene-phenotype databases. Presenting such prioritized results can potentially trigger further patient clinical examination and can lead to faster molecular diagnosis together with more appropriate patient management.

On the other hand, Exomiser failed to prioritize the correct diagnosed variants as the top 5 candidates for five (4%) patients (i.e., ranks 7, 9, 12, 39, and 42, using the DEFAULT analysis settings, Table 3). That was generally due to incompleteness or inaccuracy of the disease and phenotype databases used in the analysis. Exomiser and similar approaches should not be regarded as tools for perfect automation of the variant prioritization process but rather aids to support the lengthy, multidisciplinary effort carried out by various clinicians, scientists, and analysts to pinpoint disease-causing variants. Nevertheless, as these databases are continuously updated, prioritization is expected to improve [86]. Another limitation of Exomiser consists in its current inability of incorporating CNVs in the analysis of SNVs and indels from VCF files. CNVs have been implicated in various diseases such as autoimmune and neuropsychiatric disorders [87] and are expected to explain part of the rare disease patients without a genetic diagnosis. Future Exomiser releases will be extended to include CNV analysis functionality.

The sample size of the IRD dataset used in this analysis (i.e., 134) is among the largest real patient tested with phenotype-driven prioritization tools. Nevertheless, it would be desirable to test larger datasets on a broader range of rare Mendelian diseases to achieve more accurate and definite insights into Exomiser software performance. Indeed, Exomiser has been already adopted in many sequencing data analysis pipelines to support the identification of Mendelian disease-causing variants [88,89,90,91,92,93,94,95]. Exomiser can also efficiently process whole-genome sequencing samples (in about 5 minutes if only coding data are analyzed and in about 20 minutes for both coding and noncoding data) and is now routinely used in the variant interpretation pipeline of the 100,000 Genomes Project (https://www.genomicsengland.co.uk/) [96].

Many other HPO-driven variant prioritization tools exist [34,35,36,37,38,39,40,41,42,43,44,45,46,47,48,49,50,51,52], all primarily tested only on (different) simulated datasets with limited software performance comparison. Further work should include a systematic and exhaustive software performance comparison on both simulated and real patient large datasets. Before embarking in such task, a thorough literature review is warranted to identify the tools that could be feasibly used (e.g., freely available, with command-line access, reasonably maintained, and/or recently updated).

In conclusion, using real patient data with known molecular diagnosis, we showed how Exomiser is an effective variant prioritization tool for the integrative analysis of rare Mendelian HPO-encoded clinical phenotypes and whole-exomes. We recommend using Exomiser as an effective bioinformatics tool to support and speed up whole-exome sequencing data analysis for molecular diagnostics in rare Mendelian disease.

## Figures and Tables

**Figure 1 genes-11-00460-f001:**
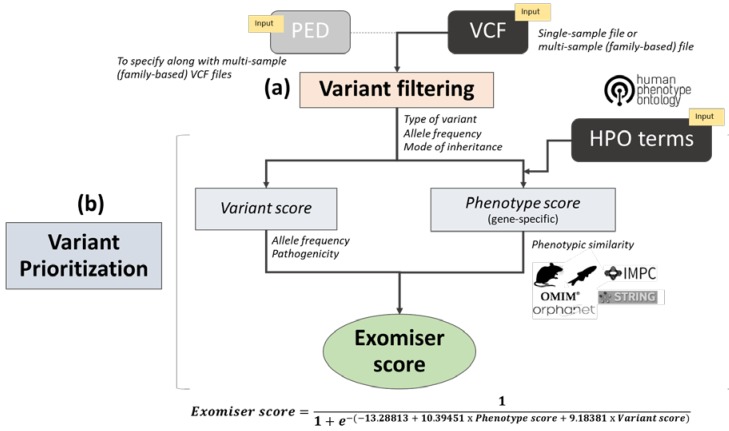
Overview of the Exomiser workflow analysis: The diagram depicts the two main steps in an Exomiser analysis: (**a**) variant filtering and (**b**) variant prioritization. A single-sample variant call format (VCF) file and corresponding list of Human Phenotype Ontology (HPO) terms are mandatory inputs. If a multi-sample VCF file (from a nuclear family) is used in the analysis, the user must provide a corresponding pedigree file. In the filtering step, variants are filtered according to type of variant, allele frequency in selected databases, and mode of inheritance as per user-defined options and values. In the prioritization step, a variant score is calculated based on allele frequency and pathogenicity as predicted by user-defined in silico algorithms, together with a gene-specific phenotype score based on the semantic similarity of the patient’s HPO terms and phenotypic annotation in known human disease gene, mouse, zebrafish and protein–protein interaction databases. Finally, the Exomiser score is obtained from the variant score and phenotypic score within a logistic regression classifier framework and used for variant prioritization. PED: Pedigree, OMIM: Online mendelian inheritance in man, IMPC: International mouse phenotyping consortium, STRING: Search tool for the retrieval of interacting genes/proteins.

**Figure 2 genes-11-00460-f002:**
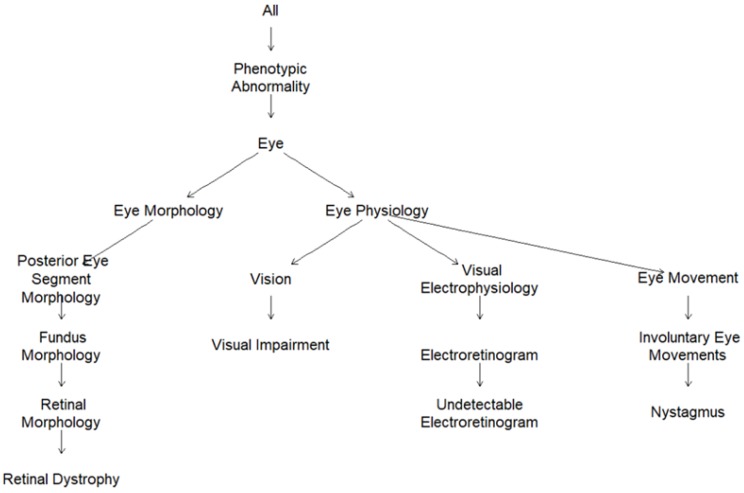
HPO graphic visualization for the HPO-encoded clinical diagnosis Leber congenital amaurosis (*Retinal dystrophy*, HP:0000556; *visual impairment*, HP:0007758, *undetectable electroretinogram*, HP:0000550; and *Nystagmus*, HP:0000639).

**Figure 3 genes-11-00460-f003:**
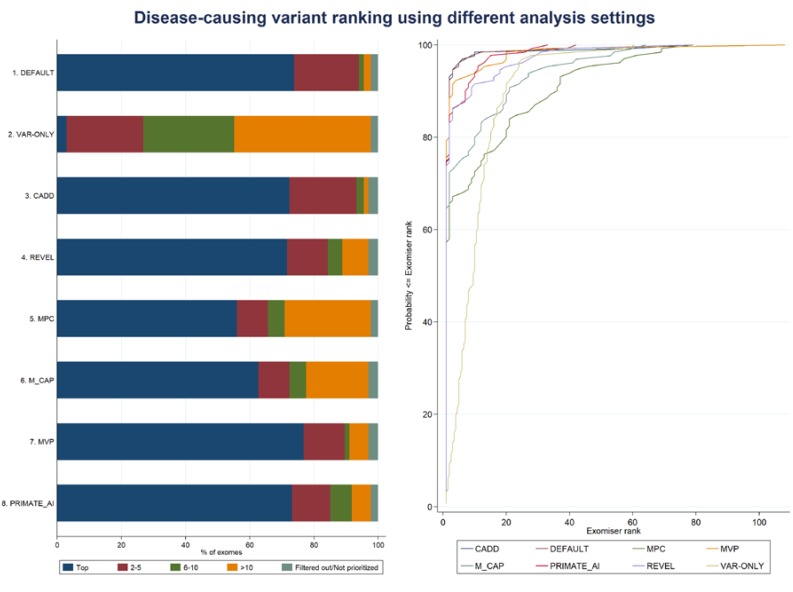
Exomiser performance on the IRD patient dataset using different analysis settings. The left-hand side panel shows the categorical percentage distribution of the disease-causing variant ranking according to five mutually exclusive disease-causing ranking bins (“Top”, “2–5”, “6–10”, “>10”, and “Filtered out/Not prioritized”) per each analysis setting. The right-hand panel shows the corresponding cumulative percentage distributions.

**Figure 4 genes-11-00460-f004:**
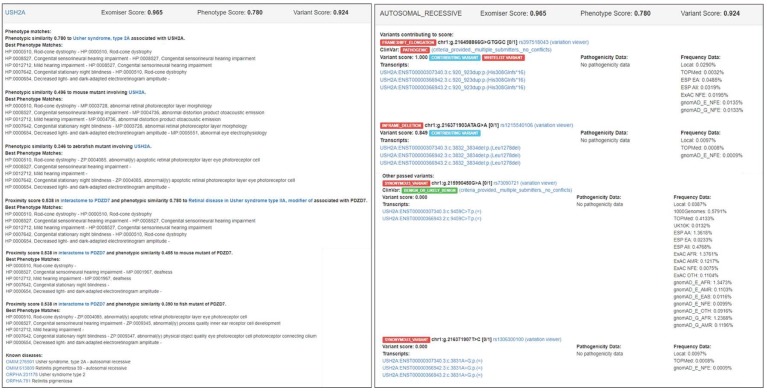
Screenshot of the Exomiser HTML output file (from the DEFAULT analysis) for patient P127 who was clinically diagnosed with Usher syndrome type II and molecularly diagnosed with frameshift elongation c.920_923dup:p.(His308Glnfs*16) and inframe deletion c.3832_3834del:p.(Leu1278del) in *USH2A* (Appendix A).

**Table 1 genes-11-00460-t001:** Frequency distribution of the clinical diagnosis in the inherited retinal disease (IRD) patient dataset.

Clinical Diagnosis ^a^	N	%
Retinitis pigmentosa (RP)	36	26.9
Leber congenital amaurosis (LCA)	25	18.7
Macular dystrophy (MD)	16	11.9
Cone-rod dystrophy (CRD)	14	10.4
Early onset retinal dystrophy (EORD)	9	6.7
Usher syndrome type II (USH2)	8	6.0
Achromatopsia (ACHM)	6	4.5
Congenital stationary night blindness (CSNB)	5	3.7
Retinal dystrophy (RD)	3	2.2
Usher syndrome type I (USH1)	2	1.5
Stargardt disease (STGD)	2	1.5
Occult macular dystrophy (OCMD)	1	0.7
Benign fleck retina (BFR)	1	0.7
Coloboma (COLOB)	1	0.7
Familial exudative vitreoretinopathy (FEVR)	1	0.7
Foveal hypoplasia (FH)	1	0.7
Myopia and deafness (Stickler syndrome) (STICKL)	1	0.7
Ocular albinism (OALB)	1	0.7
Optic atrophy (OATR)	1	0.7
**Total**	**134**	**100.0**

^a^ As assigned by a consultant ophthalmologist before performing whole-exome sequencing.

**Table 2 genes-11-00460-t002:** Frequency distribution of the genotype of the “solved” molecular diagnoses in the IRD patient dataset.

Genotype	N	%
Homozygote	72	53.7
Compound heterozygote	39	29.1
Heterozygote	13	9.7
Hemizygote ^a^	10	7.5
**Total**	**134**	**100.0**

^a^ Male patients with diagnosed variants on the X chromosome.

**Table 3 genes-11-00460-t003:** Exomiser analysis YML files (.yml) using different settings. Bold stands for analysis setting names and italics for functions.

Analysis YML File	
**1. DEFAULT** *analysis*: *genomeAssembly*: hg19 *vcf*: path-to-VCF-file *hpoIds*: [comma-separated-list-of-HPO-terms] *inheritanceModes*: { AUTOSOMAL_DOMINANT: 0.1, AUTOSOMAL_RECESSIVE_HOM_ALT: 0.5, AUTOSOMAL_RECESSIVE_COMP_HET: 2.0, X_DOMINANT: 0.1, X_RECESSIVE_HOM_ALT: 0.5, X_RECESSIVE_COMP_HET: 2.0, } *analysisMode*: PASS_ONLY *frequencySources*: [LOCAL, THOUSAND_GENOMES, TOPMED, UK10K, ESP_AFRICAN_AMERICAN, ESP_EUROPEAN_AMERICAN, ESP_ALL, EXAC_AFRICAN_INC_AFRICAN_AMERICAN, EXAC_AMERICAN, EXAC_SOUTH_ASIAN, EXAC_EAST_ASIAN, EXAC_FINNISH, EXAC_NON_FINNISH_EUROPEAN, EXAC_OTHER, GNOMAD_E_AFR, GNOMAD_E_AMR, GNOMAD_E_EAS, GNOMAD_E_FIN, GNOMAD_E_NFE, GNOMAD_E_OTH, GNOMAD_E_SAS, GNOMAD_G_AFR, GNOMAD_G_AMR, GNOMAD_G_EAS, GNOMAD_G_FIN, GNOMAD_G_NFE, GNOMAD_G_OTH, GNOMAD_G_SAS] *pathogenicitySources*: [POLYPHEN, MUTATION_TASTER, SIFT] *steps*: [ *qualityFilter*: {minQuality: 30.0} *variantEffectFilter*: { *remove*: [FIVE_PRIME_UTR_EXON_VARIANT, FIVE_PRIME_UTR_INTRON_VARIANT, THREE_PRIME_UTR_EXON_VARIANT, THREE_PRIME_UTR_INTRON_VARIANT, NON_CODING_TRANSCRIPT_EXON_VARIANT, UPSTREAM_GENE_VARIANT, INTERGENIC_VARIANT, REGULATORY_REGION_VARIANT, CODING_TRANSCRIPT_INTRON_VARIANT, NON_CODING_TRANSCRIPT_INTRON_VARIANT, DOWNSTREAM_GENE_VARIANT] }, *frequencyFilter*: {maxFrequency: 2.0}, *pathogenicityFilter*: {keepNonPathogenic: true}, *inheritanceFilter*: {}, *omimPrioritiser*: {}, *hiPhivePrioritiser*: {} ]	**2. VAR-ONLY** As per DEFAULT, but without *omimPrioritiser*: {} and *hiPhivePrioritiser*: {}
	**3. CADD** As per DEFAULT, but with *pathogenicitySources*: [CADD]
	**4. REVEL** As per DEFAULT, but with *pathogenicitySources*: [REVEL]
	**5. MPC** As per DEFAULT, but with *pathogenicitySources*: [MPC]
	**6. M_CAP** As per DEFAULT, but with *pathogenicitySources*: [M_CAP]
	**7. MVP** As per DEFAULT, but with *pathogenicitySources*: [MVP]
	**8. PRIMATE-AI** As per DEFAULT, but with *pathogenicitySources*: [PRIMATE-AI]




LOCAL is University College London exome database (UCLex) [74]. Ensembl transcript annotation was used across all the analysis settings.

**Table 4 genes-11-00460-t004:** Descriptive statistics of the Exomiser disease-causing variant ranking in the IRD patient dataset using different analysis settings.

Analysis Setting	Variants Filtered out	Variants not Prioritized ^a^	Mean Rank (SD)	Median Rank	Min Rank	Max Rank	Top Ranked, % (*N* = 134)
1. DEFAULT	2	1 (gene rank: 1)	2.1 (5.0)	1	1	42	73.9
2. VAR-ONLY	2	1 (gene rank: 30)	10.8 (9.0)	9.5	1	60.5	3.0
3. CADD	2	2 (gene ranks: 1, 2)	2.5 (8.4)	1	1	77	72.4
4. REVEL	2	2 (gene ranks: 2, 9)	3.9 (9.2)	1	1	78	71.6
5. MPC	2	1 (gene rank: 2)	10.1 (16.6)	1	1	79	56.0
6. M_CAP	2	2 (gene ranks: 2, 18)	6.8 (12.5)	1	1	64	62.7
7. MVP	2	2 (gene ranks: 2, 4)	3.1 (10.3)	1	1	108	76.9
8. PRIMATE-AI	2	1 (gene rank: 2)	2.7 (5.0)	1	1	33	73.1

^a^ In the case of patients with the disease-causing variants passing the filtering step but not being prioritized (i.e., not “contributing” to the final Exomiser score), the ranks given to the correct diagnosed gene are indicated in parentheses. The mean/median/min/max ranks in the table refer to the effective N, that is 134 − (# of patients with disease-causing variants filtered out) − (# of patients with disease-causing variants not prioritized).

**Table 5 genes-11-00460-t005:** Pairwise agreement between Exomiser disease-causing ranking results in the IRD patient dataset using different analysis settings.

DEFAULT vs. VAR-ONLY	Top	2–5	6–10	>10	Filtered out/Not Prioritized	Total		DEFAULT vs. M_CAP	Top	2–5	6–10	>10	Filtered out/Not Prioritized	Total	
Top	**4**	28	28	39		**99**	**Agreement, %**	Top	**74**	3	4	18		**99**	**Agreement, %**
2–5		**3**	9	15		**27**	7.6	2–5	*10*	**9**	1	6	1	**27**	64.6
6–10			**1**	1		**2**	**Cohen’s kappa**	6–10		*1*		1		**2**	**Cohen’s kappa**
>10		*1*		**2**		**3**	−0.013 (“poor”)	>10			*2*	**1**		**3**	0.27 (“fair”)
Filtered out/Not prioritized					**3**	**3**	**Stuart–Maxwell *P***	Filtered out/Not prioritized					**3**	**3**	Stuart–Maxwell *P*
**Total**	**4**	**32**	**38**	**57**	**3**	**134**	3.4 × 10^−22^	**Total**	**84**	**13**	**7**	**26**	**4**	**134**	8.5 × 10^−6^
**DEFAULT vs.** **CADD**	**Top**	**2–5**	**6–10**	**>10**	**Filtered** **out/Not prioritized**	**Total**		**DEFAULT vs.** **MVP**	**Top**	**2–5**	**6–10**	**>10**	**Filtered** **out/Not prioritized**	**Total**	
Top	**93**	5			1	**99**	**Agreement, %**	Top	**93**	1	1	4		**99**	**Agreement, %**
2–5	*2*	**23**	1		1	**27**	92.3	2–5	*10*	**15**		1	1	**27**	84.6
6–10	*1*		**1**			**2**	**Cohen’s kappa**	6–10		*1*		1		**2**	**Cohen’s kappa**
>10			*1*	**2**		**3**	0.80 (“substantial”)	>10			*1*	**2**		**3**	0.58 (“moderate”)
Filtered out/Not prioritized	*1*				**2**	**3**	**Stuart–Maxwell *P***	Filtered out/Not prioritized					**3**	**3**	**Stuart–Maxwell *P***
**Total**	**97**	**28**	**3**	**2**	**4**	**134**	0.818	**Total**	**103**	**17**	**2**	**8**	**4**	**134**	0.040
**DEFAULT vs.** **REVEL**	**Top**	**2–5**	**6–10**	**>10**	**Filtered** **out/Not prioritized**	**Total**		**DEFAULT vs.** **PRIMATE_AI**	**Top**	**2–5**	**6–10**	**>10**	**Filtered** **out/Not prioritized**	**Total**	
Top	**85**	4	3	7		**99**	**Agreement, %**	Top	**88**	3	5	3		**99**	**Agreement, %**
2–5	*11*	**12**	1	2	1	**27**	75.4	2–5	*10*	**12**	1	3	1	**27**	79.2
6–10		*1*		1		**2**	Cohen’s kappa	6–10		*1*	**1**			**2**	**Cohen’s kappa**
>10			*2*	**1**		**3**	0.40 (“fair”)	>10			*1*	**2**		**3**	0.48 (“moderate”)
Filtered out/Not prioritized					**3**	**3**	**Stuart–Maxwell *P***	Filtered out/Not prioritized			1		**2**	**3**	**Stuart–Maxwell *P***
**Total**	**96**	**17**	**6**	**11**	**4**	**134**	0.011	**Total**	**98**	**16**	**9**	**8**	**3**	**134**	0.011
**DEFAULT vs.** **MPC**	**Top**	**2–5**	**6–10**	**>10**	**Filtered** **out/Not prioritized**	**Total**									
Top	**66**	3	5	25		**99**	**Agreement, %**								
2–5	*9*	**9**	1	7	**1**	**27**	59.2								
6–10		*1*		1		**2**	**Cohen’s kappa**								
>10			*1*	**2**		**3**	0.24 (“fair”)								
Filtered out/Not prioritized				1	**2**	**3**	**Stuart–Maxwell *P***								
**Total**	**75**	**13**	**7**	**36**	**3**	**134**	1.0 × 10^−7^								

Table contingency counts represent number of patients at which the corresponding correct diagnosed variants were ranked by two analysis settings. DEFAULT: the Exomiser score was obtained from the variant score based on allele frequency (as defined in the filtering step, Table 3) and the original pathogenicity algorithms PolyPhen-2 [14], MutationTaster [13], and SIFT [75] using *pathogenicitySources*: [POLYPHEN, MUTATION_TASTER, SIFT], plus the gene-specific phenotype score, using *hiPhivePrioritiser*: {} and *omimPrioritiser*: {}; VAR-ONLY: as in DEFAULT analysis setting but using only the variant score, i.e., *hiPhivePrioritiser* and *omimPrioritiser* were disabled; CADD/REVEL/MPC/M_CAP/MVP/PRIMATE-AI: as in DEFAULT analysis setting using both the variant score and the phenotype score but making use of only one of the newly added pathogenicity algorithms at a time to calculate the variant score, e.g., *pathogenicitySources*: [CADD]. Agreement between two analysis settings is summarised by the percentage agreement and Cohen’s kappa and assessed via Stuart–Maxwell test. Cohen’s kappa values are interpreted according to Landis and Koch’s guidelines (i.e., κ < 0.00 as “poor” agreement, 0.00–0.20 as “slight”, 0.21–0.40 as “fair”, 0.41–0.60 as “moderate”, 0.61–0.80 as “substantial”, and 0.81–1 as “almost perfect” agreement) [76]. Counts in bold (on the diagonals) represent agreement between two analysis settings (both settings assigned the diagnosed variants the same rank); counts underlined represent when a row analysis setting performed better than a column analysis setting; and counts in *italics* represent when a row analysis setting performed worse than a column analysis setting. Blank cells represent zero counts.

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
