# Peer review of "An Improved Phenotype-Driven Tool for Rare Mendelian Variant Prioritization: Benchmarking Exomiser on Real Patient Whole-Exome Data"

_genes, 2020, doi:10.3390/genes11040460_

Round 1

Reviewer 1 Report

The authors revised and improved the manuscript. I have no further comments.

Author Response

We thank again the reviewer for their careful reading of the manuscript and their constructive remarks. 

Some minor stylistic edits have been made and are highlighted with track changes. 

Reviewer 2 Report

The manuscript by Cipriani et al makes a strong case for the utility of the Exomiser software in prioritizing genetic variants likely to be causative for rare disease.  It represents a substantial advance over previous analyses because it measures software performance using real, rather than simulated, patient data.  

While the presentation is generally clear, it could be improved by another pass of editing aimed at reducing the frequency of long run-on sentences and making as much of the text as possible accessible to non-bioinformatician readers.  A few specific suggestions follow below.

Figure 1 and associated text:  The formula for computing the Exomiser score is complex and non-intuitive.  A few sentences that describes the logic behind it and how it was derived would be helpful.

One paragraph appears to contain some missing references with the letters X, Y, and Z in brackets.  These are on lines 51, 53, 63, and 66.  This needs to be corrected.

On line 570 the gene name DHX38 is misspelled.

Author Response

The manuscript by Cipriani et al makes a strong case for the utility of the Exomiser software in prioritizing genetic variants likely to be causative for rare disease.  It represents a substantial advance over previous analyses because it measures software performance using real, rather than simulated, patient data. 

While the presentation is generally clear, it could be improved by another pass of editing aimed at reducing the frequency of long run-on sentences and making as much of the text as possible accessible to non-bioinformatician readers.  A few specific suggestions follow below.

Figure 1 and associated text:  The formula for computing the Exomiser score is complex and non-intuitive.  A few sentences that describes the logic behind it and how it was derived would be helpful.

Authors’ reply

We thank again the reviewer for their careful reading of the manuscript and their constructive remarks.

We welcomed their suggestions and made many stylistic edits throughout the text (lines 48, 57-63, 76, 115-117, 547, 597-598); as to the Exomiser formula, although it may look complex at first sight, it is derived from a very standard logistic regression classifier and we are confident most readers would follow that. However, we made a more explicit reference to the logistic regression classifier in the main text and Figure 1 caption (lines 290-293, 140).

Reviewer 2

One paragraph appears to contain some missing references with the letters X, Y, and Z in brackets.  These are on lines 51, 53, 63, and 66.  This needs to be corrected.

Authors’ reply

This must have been missed out by the editorial team. We had indicated the new X, Y, Z references in the comment field. We have now added the full references in the main text and added a note for the editorial team in the cover letter.

Reviewer 2

On line 570 the gene name DHX38 is misspelled.

Authors’ reply

Thanks for spotting this typo. It has been amended.

Reviewer 3 Report

Please find enclosed some comments for your consideration

Thanks, 

Author Response

A reply to the third reviewer's comments is included in the cover letter. 

Round 2

Reviewer 3 Report

The paper is fine and ready for publication

This manuscript is a resubmission of an earlier submission. The following is a list of the peer review reports and author responses from that submission. Reviewer 1 agreed to review again, reviewer 2 changed. 

Round 1

Reviewer 1 Report

Cipriani et al present a very compelling article describing the utility of the latest version of Exomiser (v 12.0.1) in identifying causative disease variants in a large inherited retinal disease cohort.  The authors analyzed Exomiser performance using exome data from 134 'solved' cases. The manuscript is exceedingly well-written, presents the data in a clear and organized fashion, and appropriately addresses the limitations of Exomiser (does not currently analyze/annotate CNVs, not as informative in identifying novel disease genes) and limitations of study design/ focusing on this particular study cohort (analysis should be replicated in additional cohorts with other rare diseases).

My main comment/area to address was it is not clear to me how many of the 134 cases included trio analysis vs singleton analysis of exomes (ie, how were compound heterozygous variants identified/confirmed to be compound het if only singleton exome was completed). I would appreciate more details in the manuscript regarding whether trio or singleton exome testing was completed for these 134 cases.

I believe this manuscript is of very high interest to clinical and molecular geneticists who focus on identifying causative disease variants for rare disease.

Author Response

Authors’ note to both reviewers:
In a careful revision of the analysis, we identified one more patient that belongs to the category ‘Filtered out’ (i.e., the diagnosed variant was erroneously called heterozygous in the VCF file and did not pass the Exomiser mode of inheritance filtering). Also, for a total of 3 other patients, the correct diagnosed variants did pass the filtering step, but were not prioritized as ‘contributing’ variants to the Exomiser score, being not the variants with the highest variant score in the gene. For this scenario, we created the category ‘Not prioritized’ and pulled it together with the ‘Filtered out’ category. All corresponding results in the main text, tables and figures have been updated with very minor changes and no changes in the overall results and interpretation.

Comments and Suggestions for Authors – Reviewer 1

Cipriani et al present a very compelling article describing the utility of the latest version of Exomiser (v 12.0.1) in identifying causative disease variants in a large inherited retinal disease cohort. The authors analyzed Exomiser performance using exome data from 134 'solved' cases. The manuscript is exceedingly well-written, presents the data in a clear and organized fashion, and appropriately addresses the limitations of Exomiser (does not currently analyze/annotate CNVs, not as informative in identifying novel disease genes) and limitations of study design/ focusing on this particular study cohort (analysis should be replicated in additional cohorts with other rare diseases).

Reviewer’s Point 1.
My main comment/area to address was it is not clear to me how many of the 134 cases included trio analysis vs singleton analysis of exomes (ie, how were compound heterozygous variants identified/confirmed to be compound het if only singleton exome was completed). I would appreciate more details in the manuscript regarding whether trio or singleton exome testing was completed for these 134 cases.
I believe this manuscript is of very high interest to clinical and molecular geneticists who focus on identifying causative disease variants for rare disease.

Reviewer’s Reply 1.
We thank the reviewer for their positive feedback.
As to their main comment – the exome data presented in the manuscript were accumulated as part of a number of sequencing studies at the UCL Institute of Ophthalmology and Moorfields Eye Hospital over approximately 4 years (2011-2015). The vast majority of patients were sequenced as singletons, expect for 6 probands where one or both parents were also sequenced. Where possible, all variants, including compound heterozygous variants, were validated in an accredited clinical diagnostic laboratory. Familial segregation analysis was performed where appropriate to reach a diagnostic conclusion.
Methods have been edited accordingly (lines 150-172).

Reviewer 2 Report

Reviewer Blind Comments to Author

Remarks on the manuscript entitled "An Improved Phenotype-Driven Tool for Rare Mendelian Variant Prioritization: Benchmarking Exomiser on Real Patient Whole-Exome Data”.

The authors present the latest release of the software tool Exomiser for phenotype-driven variant filtering and prioritization, describing Exomiser’ main newly added features, as well as show the software performance on a whole-exome dataset of 134 individuals affected by rare inherited retinal disease who had previously received a molecular diagnosis based on the analysis of their NGS data. This is an interesting and well-written work with considerable clinical/scientific impact but there are some points that have to be considered and revised:

- Although the authors mention that they used 134 diagnosed patients with rare retinal disorders, the selection of these individuals is not clear enough, i.e. it is unclear whether these patients were consecutive unbiased cases; otherwise the selection of cases could have been biased towards such in which Exomiser performed well or variant interpretation was less challenging.

- Although the authors state that “Exomiser now allows both coding and non-coding variant prioritization from whole-genome sequencing [WGS] data” (lines 107, 108), only whole-exome sequencing (WES) data were used in this study. As WES yields a substantially lower number of to-be-filtered variants compared to WGS and as WGS is the future of sequencing (at least in rare diseases), it would be timely to extend Exomiser’s performance analysis to WGS data as well (~3 mio variants), i.e. I am wondering whether the authors can demonstrate or predict  Exomiser’s performance for WGS data.

For instance, the intronic variant FBN2 c.3974-26T>G p.(Asn1327_Val1368del) (doi.org/10.1086/515472) exemplifies several caveats in variant filtering and interpretation, which may be missed by Exomiser’s default settings and thus may be, at least, discussed by the authors. Indeed, (deep) intronic variants may also be disease-causing, particularly such that affect splicing, which might be missed using WES (regardless of the interpretation platform). Although the recent implementation of ClinVar in Exomiser is a great feature, this intronic FBN2 variant is most likely missed by Exomiser (unless deep intronic variants are allowed by the variantEffectFiler, which is not the default setting). I am wondering whether or not Exomiser’s variant prioritization can consider variants described in HGMD and/or Pubmed but not listed in ClinVar.

- The default settings of the frequency filters are set to 0.1% in population-based reference datasets e.g. for autosomal-dominant disorders, but without stating a reason for this specific value. The authors should specify the reason, also considering that implementing very low frequency cutoffs may lead to false-negative results, as recently discussed (cf. doi.org/10.1111/cge.13640).

- Lines 47-49: The authors state that the “main reason why a disorder remains unsolved after undergoing next-generation sequencing is the complexity of the interpretation of the wealth of variants found”. This is somehow misleading and incomplete as the reason of unsolved cases is not only due to the high number of detected variants but also due to the limitations of short-read WES/WGS and due to the limited knowledge on disease-gene associations and gene functions (cf. Pubmed: 29206278).

- Lines 76-80: The authors mention several phenotype-driven interpretation tools but I miss the AI-based platform moon (diploid.com/moon), which, to my knowledge, is the fastest annotation/interpretation platform currently available for both WES- and WGS-based variant interpretation.

- Lines 562-572: The authors discuss the limitations of Exomiser but I miss information on the ability/limitation of Exomiser to interpret/prioritize mtDNA variation.

- In Table 5, comparison DEFAULT vs. CADD: Should not the total number of filtered out variants be “1” instead of “2”?

- Figure 4, legend: Revise “deletion p.(Leu1278del)” to “deletion c.3832_3834del:p.(Leu1278del)” and use “USH2A” in italic.Supplementary Tables S3 and S4: Some of the text is cut off at the right boarder of the table (in my MS Word version). I recommend that the authors provide the supplementary file as pdf instead of docx (to avoid any display difficulties).

Author Response

Remarks on the manuscript entitled "An Improved Phenotype-Driven Tool for Rare Mendelian Variant Prioritization: Benchmarking Exomiser on Real Patient Whole-Exome Data”.
The authors present the latest release of the software tool Exomiser for phenotype-driven variant filtering and prioritization, describing Exomiser’ main newly added features, as well as show the software performance on a whole-exome dataset of 134 individuals affected by rare inherited retinal disease who had previously received a molecular diagnosis based on the analysis of their NGS data. This is an interesting and well-written work with considerable clinical/scientific impact but there are some points that have to be considered and revised:

Reviewer’s Point 1.
Although the authors mention that they used 134 diagnosed patients with rare retinal disorders, the selection of these individuals is not clear enough, i.e. it is unclear whether these patients were consecutive unbiased cases; otherwise the selection of cases could have been biased towards such in which Exomiser performed well or variant interpretation was less challenging.

Reviewer’s Reply 1.
This has been addressed by editing substantially the relevant methods section “2.1 Real patient whole-exome dataset with known molecular diagnosis”. We have added many details on the overall patient cohort and the extracted whole-exome dataset with known molecular diagnosis (lines 150-172):

“We evaluated Exomiser software performance using a real patient whole-exome dataset with known molecular diagnosis obtained from a number of next-generation sequencing studies that were conducted at Moorfields Eye Hospital and the UCL Institute of Ophthalmology (London, United Kingdom) approximately over 4 years (2011-2015). The dataset was extracted from the overall patient cohort of 340 individuals that had been diagnosed with IRD by a consultant ophthalmologist and had undergone WES. All patients had been sequenced as singletons, expect for 6 probands where one or both parents were also sequenced. The patient cohort was enriched for rare genetic causes of IRD (many of which have been published over the years). Cases were selected for their likelihood of harbouring a novel discovery based on many features, including family consanguinity/number of affected siblings in recessive families, prior negative testing for the ASPER chip, candidate gene sequencing or the Manchester 105-gene diagnostic panel, unusual phenotype/clinical history, prior linkage/SNP-homozygosity mapping, absence of typical clinical features of ABCA4-retinopathy or other (near) monogenic disease (e.g., X-linked retinoschisis, RS1 or choroideremia, CHM). A total of 134 IRD patients that had received a molecular diagnosis (i.e., solved patients) at the time of the data extraction based on single nucleotide variants (SNVs) and/or indels (as present in their respective VCF files) were included in the Exomiser software performance assessment. Twelve additional patients with a molecular diagnosis that involved CNVs could not be used due to the current software inability of incorporating CNVs into the analysis. The molecular diagnosis of the IRD patients had been based on the analysis and interpretation of the patient WES data performed by various clinicians, scientists and analysts, without using Exomiser. Where possible, diagnosed variants, including compound heterozygous variants, had been validated in an accredited clinical diagnostic laboratory. Familial segregation analysis had been performed where appropriate to reach a diagnostic conclusion.”

Reviewer’s Point 2.
Although the authors state that “Exomiser now allows both coding and non-coding variant prioritization from whole-genome sequencing [WGS] data” (lines 107, 108), only whole-exome sequencing (WES) data were used in this study. As WES yields a substantially lower number of to-be-filtered variants compared to WGS and as WGS is the future of sequencing (at least in rare diseases), it would be timely to extend Exomiser’s performance analysis to WGS data as well (~3 mio variants), i.e. I am wondering whether the authors can demonstrate or predict Exomiser’s performance for WGS data.

Reviewer’s Reply 2.
As noted in the README file of the software documentation (https://github.com/exomiser/Exomiser), for a whole-exome analysis of a 30,000 variant sample, 4GB RAM are usually required and the analysis can take 1-2 minutes; for a whole-genome analysis of a 4,400,000 variant sample, 12GB RAM should suffice and the analysis can take up to 20 minutes; this goes down to 4-5 minutes if only coding data (from a whole-genome) are analysed. The availability of molecularly solved patient data with non-coding variants is still limited; as the reviewer mentioned, this is expected to change in the future as whole-genome sequencing (WGS) becomes standard in rare disease sequencing. Thus, software performance statistics on WGS patient data should become available too.
Exomiser is now routinely used as part of the WGS diagnostic pipeline for the 100,000 Genomes Project (100KGP) and the subsequent NHS Genomic Medicine Service (see the rare disease guide at https://www.genomicsengland.co.uk/?wpdmdl=15664), and the 100KGP is expected to describe the effective performance of Exomiser for WGS interpretation in 2020.
This was already mentioned in the discussion and is now edited as follows: “Indeed, Exomiser has been already adopted in many sequencing data analysis pipelines to support the identification of Mendelian disease-causing variants [83-90]. Exomiser can also efficiently process whole-genome sequencing samples (in about 5 minutes if only coding data are analyzed; about 20 minutes for both coding and non-coding data) and is now routinely used in the variant interpretation pipeline of the 100,000 Genomes Project (https://www.genomicsengland.co.uk/) [91].” (lines 599-602).

Reviewer’s Point 3.
For instance, the intronic variant FBN2 c.3974-26T>G p.(Asn1327_Val1368del) (doi.org/10.1086/515472) exemplifies several caveats in variant filtering and interpretation, which may be missed by Exomiser’s default settings and thus may be, at least, discussed by the authors. Indeed, (deep) intronic variants may also be disease-causing, particularly such that affect splicing, which might be missed using WES (regardless of the interpretation platform). Although the recent implementation of ClinVar in Exomiser is a great feature, this intronic FBN2 variant is most likely missed by Exomiser (unless deep intronic variants are allowed by the variantEffectFiler, which is not the default setting). I am wondering whether or not Exomiser’s variant prioritization can consider variants described in HGMD and/or Pubmed but not listed in ClinVar.

Reviewer’s Reply 3.
We thank the reviewer for providing this example. Although the intronic variant FBN2 c.3974-26T>G p.(Asn1327_Val1368del) would be missed by the Exomiser’s default coding-based ‘variantEffectFilter’ settings and the ClinVar-based default whitelist, the user has indeed the option to edit the variant whitelist (that is an openly available standard tabix indexed file) and incorporate any variants they want to keep bypassing any filtering settings. We have edited the manuscript accordingly (lines 266-268): “Indeed, the user can edit the variant whitelist (i.e., an openly available tabix indexed file) by adding any variants they want to keep, bypassing any filtering settings.”.

Reviewer’s Point 4.
The default settings of the frequency filters are set to 0.1% in population-based reference datasets e.g. for autosomal-dominant disorders, but without stating a reason for this specific value. The authors should specify the reason, also considering that implementing very low frequency cutoffs may lead to false-negative results, as recently discussed (cf. doi.org/10.1111/cge.13640).

Reviewer’s Reply 4.
First, apologies - we noted a typo in Table 3 in the DEFAULT settings: the cut-off that we used both for AUTOSOMAL_RECESSIVE_HOM_ALT and X_RECESSIVE_HOM_ALT was 0.5 (%), and not 0.1(%). The table has now been amended.
We agree with the reviewer that very low frequency cut-offs may lead to false negatives. We used cut-offs 0.5% and 0.1% for autosomal-recessive and autosomal-dominant, respectively, in order to match the filtering criteria of the bioinformatics pipeline used at UCL/MEH to solve the cases. We have now added this detail in the methods section (lines 300-302).
Also, we added the following text to the introduction: “On the other hand, although selecting variants that are extremely rare or absent in population-based reference datasets may be helpful in a first step of the analysis, using very low frequency cut-offs may remove clinically relevant variants and lead to false negative findings, especially in hard to solve cases [doi.org/10.1111/cge.13640]. More complex statistical frameworks that account for disease prevalence, genetic and allelic heterogeneity, inheritance mode, penetrance and sampling variance in reference datasets have been recently suggested for a more effective frequency-based variant filtering [doi.org/10.1111/cge.13640, doi.org/10.1038/gim.2017.26].” (lines 60-66).

Reviewer’s Point 5.
Lines 47-49: The authors state that the “main reason why a disorder remains unsolved after undergoing next-generation sequencing is the complexity of the interpretation of the wealth of variants found”. This is somehow misleading and incomplete as the reason of unsolved cases is not only due to the high number of detected variants but also due to the limitations of short-read WES/WGS and due to the limited knowledge on disease-gene associations and gene functions (cf. Pubmed: 29206278).

Reviewer’s Reply 5.
We agree with the reviewer and thank them for highlighting this interesting reference. We have edited the introduction as follows: “Several limitations of high-throughput sequencing still exist that may impact the diagnostic yield, including the yet incomplete coverage affecting especially short-read sequencing, the non-existence of a de facto standard calling algorithm for copy-number variants (CNVs) or the challenges in filtering and interpreting short tandem repeats with no known disease association [DOI: 10.1111/cge.13190]. Another major reason why a disorder remains unsolved after undergoing next-generation sequencing is the complexity of the interpretation of the wealth of variants found, which is further hindered by the incomplete knowledge on gene functions [DOI: 10.1111/cge.13190].” (lines 47-54).

Reviewer’s Point 6.
Lines 76-80: The authors mention several phenotype-driven interpretation tools but I miss the AI-based platform moon (diploid.com/moon), which, to my knowledge, is the fastest annotation/interpretation platform currently available for both WES- and WGS-based variant interpretation.

Reviewer’s Reply 6.
We respectively disagree with the reviewer: Moon can be seemingly fast, but, to our knowledge, an exhaustive software performance comparison has never been performed. Furthermore, the performance statistics discussed on the Moon website and in their white paper are based on a very limited number of samples (i.e. “30 real-life single exome cases”) and the comparison with Exomiser is based on an outdated Exomiser version (i.e. version 7.2.3; data v7.2.1). More importantly, many commercially available tools must exist, but citing commercially available software is beyond the scope of our analysis.
To avoid any confusion, we added the specification ‘freely available’ in two sentences in the introduction: “A growing number of in silico (freely available) computational tools have been developed […]” (line 67) and “In order to overcome these difficulties, another generation of (freely available) tools have been developed […]” (line 87). In the discussion, we also say: “Further work should include a systematic and exhaustive software performance comparison on both simulated and real patient large datasets. Before embarking in such task, a thorough literature review is warranted to identify the tools that could be feasibly used (e.g., with freely available, command-line access, reasonably maintained and/or recently updated).”

Reviewer’s Point 7.
Lines 562-572: The authors discuss the limitations of Exomiser but I miss information on the ability/limitation of Exomiser to interpret/prioritize mtDNA variation.

Reviewer’s Reply 7.
Assuming mitochondrial variants are called in the input VCF they will be interpreted in the same manner as nuclear variants using exactly the same sources of pathogenicity, allele frequency and disease associations. We regularly use this as part of the 100,000 Genomes Project pipeline and detect mtDNA diagnoses.